# Awareness Regarding Antimicrobial Resistance and Antibiotic Prescribing Behavior among Physicians: Results from a Nationwide Cross-Sectional Survey in India

**DOI:** 10.3390/antibiotics12101496

**Published:** 2023-09-29

**Authors:** Niti Mittal, Parul Goel, Kapil Goel, Rashmi Sharma, Bhola Nath, Surjit Singh, Pugazhenthan Thangaraju, Rakesh Mittal, Prasanna Mithra, Rajesh Sahu, Raman P. Priyadarshini, Nikita Sharma, Star Pala, Suneel Kumar Rohilla, Jyoti Kaushal, Sanjit Sah, Sarvesh Rustagi, Ranjit Sah, Joshuan J. Barboza

**Affiliations:** 1Department of Pharmacology, Pt. B. D. Sharma Postgraduate Institute of Medical Sciences, Rohtak 124001, India; drnitimittal@uhsr.ac.in (N.M.); rakeshmittal.pgims@uhsr.ac.in (R.M.); suneelrohilla.pgims@uhsr.ac.in (S.K.R.); drjyotikaushal@uhsr.ac.in (J.K.); 2Department of Biochemistry, Shri Atal Bihari Vajpayee Government Medical College, Chhainsa, Faridabad 121004, India; parul006@gmail.com; 3Department of Community Medicine and School of Public Health, Postgraduate Institute of Medical Education and Research, Chandigarh 160012, India; 4Department of Community Medicine, GMERS Medical College Sola, Ahmedabad 380060, India; drrashmi_psm@yahoo.com; 5Department of Community and Family Medicine, All India Institute of Medical Sciences, Raebareli 229405, India; bholanath.2001@aiimsrbl.edu.in; 6Department of Pharmacology, All India Institute of Medical Sciences, Jodhpur 342001, India; singhs@aiimsjodhpur.edu.in; 7Department of Pharmacology, All India Institute of Medical Sciences, Raipur 492010, India; drpugal@aiimsraipur.edu.in; 8Department of Biochemistry, All India Institute of Medical Sciences, Deoghar 814152, India; kahkasha.biochemistry@aiimsdeoghar.edu.in; 9Department of Community Medicine, Kasturba Medical College, Mangalore, Manipal Academy of Higher Education, Manipal 575001, India; prasanna.mithra@manipal.edu; 10Department of Community Medicine, Armed Forces Medical College, Pune 411040, India; rajeshsahu.024m@gov.in; 11Department of Pharmacology, Jawaharlal Institute of Postgraduate Medical Education and Research, Karaikal 609602, India; kd0017@jipmer.ac.in; 12Department of Community and Family Medicine, All India Institute of Medical Sciences, Bilaspur 174037, India; dr.nikita.cfm@aiimsbilaspur.edu.in; 13Department of Community Medicine, NEIGRIHMS, Shillong 793018, India; star.pala@gov.in; 14Global Consortium for Public Health and Research, Datta Meghe Institute of Higher Education and Research, Jawaharlal Nehru Medical College, Wardha 442001, India; sanjitsah101@gmail.com; 15School of Applied and Life Sciences, Uttaranchal University, Dehradun 248007, India; sarveshrustagi@uumail.com; 16Tribhuvan University Teaching Hospital, Kathmandu 46000, Nepal; ranjitsah@iom.edu.np; 17School of Medicine, César Vallejo University, Trujillo 13007, Peru

**Keywords:** antimicrobial resistance, stewardship, antibiograms, prescribing practices

## Abstract

(1) Background: Understanding the physicians’ knowledge, attitudes, and antimicrobial prescribing behavior is a crucial step towards designing strategies for the optimal use of these agents. (2) Methods: A cross-sectional online survey was conducted among clinicians across India between May and July 2022 using a self-administered questionnaire in English comprising 35 questions pertaining to demographic characteristics, knowledge, attitude, and practices domains. (3) Results: A total of 544 responses were received from 710 physicians contacted. Sixty percent of participants were males, with mean age of 34.7 years. Mean ± Standard Deviation scores for knowledge, attitude, and practices domains were 8 ± 1.6, 20.2 ± 3.5, and 15.3 ± 2.1, respectively. Higher scores were associated with basic [odds ratio (95% Confidence Interval), *p* value: 2.95 (1.21, 7.2), 0.02], medical and allied sciences [2.71 (1.09, 6.67), 0.03], and central zone [3.75 (1.39, 10.12), 0.009]. A substantial proportion of dissatisfactory responses were found regarding hospital antibiograms, antibiotics effective against anaerobes, WHO AWaRe (access, watch, and reserve) classification of antibiotics, and the role of infection prevention and control (IPC) measures in the containment of antimicrobial resistance (AMR). (4) Conclusions: There is a need to sensitize and educate clinicians on various issues related to antimicrobial use, such as antibiograms, double anaerobic cover, IPC practices, and guideline-based recommendations, to curb the AMR pandemic.

## 1. Introduction

Antimicrobial resistance (AMR) was attributed to more than 1.2 million deaths globally in 2019, according to the landmark GRAM (Global Research on Antimicrobial Resistance) study [1]. Indiscriminate use of antimicrobials is the major driving force for the extensive emergence and spread of multi-drug-resistant (MDR) and extensively drug-resistant (XDR) pathogens in India. [2,3]. Antimicrobials’ consumption witnessed a startling rise globally (65%) as well as in India (103%) between 2000 and 2015 [4]. Such serious affairs regarding rampant antimicrobial use and the alarming rise in AMR have stemmed numerous international and national initiatives [5,6,7,8,9], India being no exception [10,11,12]. “Global Action Plan on Antimicrobial Resistance (GAP-AMR)” was launched by World Health Organization (WHO) in 2015 [5], followed by India’s “National Action Plan on Antimicrobial Resistance” in 2017 (2017–2021) [10] to fight the AMR pandemic. A few strategic priorities listed under NAP-AMR include improving awareness and understanding of AMR amongst all stakeholders through effective information, education, and communication (IEC) resources; and optimizing the use of antimicrobials through strengthening antimicrobial stewardship programs (ASPs) in healthcare [10]. Besides these, WHO AWaRe (access, watch, and reserve) classification of antibiotics is an effective tool towards tackling the menace of AMR and promoting ASPs, especially in Low and Middle-Income Countries (LMICs) [11]. According to this classification, “access” group of antibiotics are the agents, which are generally narrow spectrum, are effective against common pathogens, and have low resistance potential; “watch” group encompasses antibiotics with higher resistance potential and highest priority agents among the ‘critically important antimicrobials for human medicine, and “reserve” group includes antibiotics, which should be reserved for infections due to multi-drug-resistant organisms. The global target of the “access” group to comprise at least 60% of total antibiotic consumption being Access group antibiotics as part of effective strategy has been defined by WHO as part of common stewardship interventions [13].

Behavioral characteristics of both physicians and patients contribute to unnecessary antibiotic prescribing patterns as prescribers’ practices are usually influenced by patient expectations and requests. There is a need to understand various factors governing the healthcare professionals’ prescribing behaviors in order to implement strategies at national and international levels to contain AMR. In this direction, surveys conducted among physicians in developed nations [14,15,16,17] and LMICs [18,19,20], in particular, highlight the fact that the non-judicious antimicrobial prescribing practices among prescribers are often attributable to lacunae in knowledge about AMR and optimal use of antimicrobials, including antibiotics. A few studies have emphasized the failure of adequate translation of the knowledge of AMR into a reduction in prescribing and dispensing of antimicrobials [7,21,22,23]. An online survey-based study of physicians from seven countries in the Asia Pacific region reported that physicians in India to be high prescribers [24]. Thakolkaran et al. conducted a KAP (knowledge, attitude, and practices) study among 350 physicians in South India and reported that the physician’s knowledge of resistance patterns of common bacteria was related to receiving periodic updates on resistance patterns of bacteria and participation in courses on antibiotics [22]. Another study reported high scores in knowledge and attitude scores among physicians in West Bengal, though the participants had poor performance in the practices domain. Although many doctors exhibited knowledge with respect to indications of antibiotics. However, over 87% reported prescribing antibiotics for viral infections [25]. In another survey involving 539 intensivists across India, Gupte et al. reported high levels of variability in the prescription patterns, advocating the need for antibiotic stewardship to standardize antibiotic prescriptions not only for efficacy but also to reduce the burden of multiple drug resistance [26]. A few other studies also reported the knowledge of AMR and antibiotic prescribing patterns among physicians in different settings in India [27,28].

Most of the earlier studies in India were conducted at local or regional levels involving different subsets of physicians and hence fail to provide a comprehensive picture regarding physicians’ knowledge, perceptions, and prescribing practices across the country and variations, if any, among different zones or physicians attributes. Hence, this survey was designed to provide an up-to-date estimate of the knowledge, attitude, and antibiotic prescribing behavior of physicians across India. The survey findings would inform research and guide the development of strategies, interventions, and policies for the containment of AMR in India.

## 2. Methods

### 2.1. Study Design

A cross-sectional study was conducted among allopathy physicians from government and private sectors across India between May and July 2022.

### 2.2. Sample Size Calculation

The sample size for the present study was based on the proportion of the respondents perceiving AMR to be a problem in their own setting. Assuming this proportion to be 50% (a conservative choice since there are no nationwide data in this regard), the minimum required sample size was estimated to be 384 with a 50% proportion rate, 5% absolute precision, and 95% confidence level using the formula:

Sample size n = [DEFF × Np(1 − p)]/ [(d2/Z21 − α/2 × (N − 1) + p × (1 − p)] [29].

The design effect was kept at 1.0. Considering the 25% non-response rate, the required sample size was 512. We recruited a total of 544 physicians out of 710 physicians contacted from May to July 2022, of which 432 physicians were from the Government sector, and 112 were from the private sector.

### 2.3. Recruitment of Study Subjects

Physicians (N = 544) were recruited from an online survey via a self-reported questionnaire (Appendix A) through social media like WhatsApp, Facebook, and Twitter. For the survey, the country (India) was divided into 5 zones—North, East, West, South, and Central zone. Physicians who have at least completed MBBS (Bachelor of Medicine and Bachelor of Surgery) and who are currently residing in India were included in the study. The questionnaire in English language, prepared in Google Forms, was sent to physicians across India by sharing links through contacts of physicians. Thus, primarily, a convenient sampling technique was used. Further, the snowball sampling technique was used for enlisting additional participants, where the invited participants were requested to pass on the invitations to their peers. The initial part of the questionnaire included the consent to participate in the study. After consent was provided, the next questions appeared in the questionnaire, and the participants filled in their responses for each question.

### 2.4. Study Questionnaire

A semi-structured, self-administered, and pre-tested questionnaire in English language was used as a data collection tool. The questionnaire comprised 35 questions grouped into 4 sections: Section 1 (9 questions) on socio-demographic characteristics, such as name, age, gender, highest educational qualification, designation, department, affiliation, years of practice, and residence; Section 2 (11 questions) on knowledge of antibiotic use and resistance; Section 3 (6 questions) on the attitude of physicians toward antibiotic prescription; and Section 4 (9 questions) on physician antibiotic prescribing behaviors in regular practice. The questionnaire was prepared after a literature review of similar studies. The preliminary draft of the questionnaire was reviewed by five expert researchers in the fields of clinical pharmacology, infectious diseases, epidemiology, public health, and internal medicine to identify ambiguity and content validity. In a pilot study, this questionnaire was then pre-tested among 20 physicians who were not part of the study to assess its duration, clarity, sequence, and feasibility. Necessary modifications were made before sending out the final questionnaires to respondents.

Participants’ knowledge of antibiotic use and resistance was assessed by a set of 11 questions covering factors promoting AMR, antibiograms, WHO AWaRe classification of antibiotics, agents effective against anaerobes, and intravenous to oral switch of antibiotics. A score of 1 was given for correct responses, while each wrong or do not know answer was scored as 0. One question (as per your knowledge, which is the most prescribed antibiotic in the COVID-19 pandemic) was not included in scoring, while a score of 0 to 3 was given for one multiple-choice question with three correct options; hence, the overall knowledge score ranged from 0 to 12.

The attitude was assessed by a set of 6 positive and negative attitude questions in the form of yes/no/don’t know (do you think that your antibiotic prescribing behavior has an impact on the development of antibiotic resistance in your region?; scored 0 or 1), multiple choice questions [who among the following can play a key role in addressing the issue of antibiotic resistance (scored 1 to 7); which of the following strategies can help in addressing the issue of antibiotic resistance (scored 1 to 4)] and on five-point Likert’s scale (strongly agree/agree/neither agree nor disagree/disagree/strongly disagree; scored 1 to 5). Overall attitude scores ranged from 5 to 27; the higher the score, the more positive the attitude.

Practice was assessed by 9 questions: on four- point Likert’s scale (always/sometimes/rarely/never; scored 1 to 4), yes/no/don’t know (do you ever employ delayed/back-up antibiotic prescribing in your clinical practice) and multiple-choice questions (which sources of information do you mostly use while making decisions on antibiotic prescription, which factor/s mostly determine your choice of antibiotics; not included in scoring). Moreover, the participants were asked if they had attended any trainings/conferences to update their knowledge of antibiotic use during the last 12 months. Overall practice scores ranged from 5 to 21, with higher scores indicating good practice.

Using modified Bloom’s cut-off point, the percentage knowledge, attitude, and practice scores were grouped into good (scores between 80 and 100%), average (scores between 50 and 79%), and poor (less than 50%) [30].

### 2.5. Statistical Analysis

Through Google form, the data were captured in Microsoft Excel 2021 and were analyzed using Statistical Package for Social Sciences [SPSS (Trial v. 28)]. Appropriate tables and graphs were prepared, and inferences were drawn by applying descriptive statistics. Gender was coded as a binary variable (male and female). The education qualification was categorized into three groups (MBBS, MD/MS/Diploma/DNB, and DM/MCh). The type of work setting was categorized into two groups: Government and private organizations, and the level of setting were categorized into three groups: primary, secondary, and tertiary care hospitals. Years of practice were categorized into four groups: <5, 5–10, 11–20, >20 years. The geographical region of India was grouped into five zones: North, South, East (including Northeast), West, and Central. A logistic regression analysis was conducted to determine predictors of aggregate knowledge, attitude, and practice scores among study participants. A *p* < 0.05 was considered statistically significant.

### 2.6. Ethical Consideration

The study was approved by the Institutional Ethics Committee of the Post Graduate Institute of Medical Sciences (PGIMS), Rohtak, India (BREC/22/22 dated 19.04.2022). An online informed consent was obtained from all the participants at the start of the survey. Only those who provided the informed consent were taken to the questionnaire web page to initiate the survey. Steps were taken to ensure confidentiality and privacy of the information provided by the participants, and only de-identified data were used.

## 3. Results

Out of a total of 710 physicians contacted from five geographical zones of India, 544 responded (non-response rate of 23.8%), and hence, 544 responses were included in the analysis.

### 3.1. Demographic Characteristics

Table 1 depicts the demographic profile of study participants. The mean age of the participants was 34.7 years, and approximately 60 percent were males. The majority had postgraduate qualifications (364; 66.9%) and worked in government settings (79.4%). The respondents belonged to different levels of settings, the bulk being tertiary care (83.8%). It was observed that around 45% (244) and 27% (149) of respondents had more than 5 and 10 years of experience, respectively.

### 3.2. Knowledge Domain of the Questionnaire

The mean (SD) score in the knowledge domain was 8 (1.6), with the majority of the participants having average (330; 60.7%) and good (208; 38.2%) scores. An overwhelmingly high proportion of the respondents were of the opinion that indiscriminate use of antibiotics (534; 98.2%) and use of broad-spectrum agents (521; 95.8%) contribute to AMR. Limited access to essential antibiotics as a contributing factor to AMR emergence was believed by around half of the respondents (51%). More than 50% of respondents were not familiar with WHO AWaRe classification of antibiotics. Although the majority felt the importance of hospital antibiograms in guiding empiric antibiotic therapy (85.7%), it was not clear to more than 60% that antibiograms from different hospitals are usually dissimilar. Knowledge with respect to agent/s effectiveness against anaerobes was not satisfactory, with only 171 (31.5%) correct responses. There was variation among participants with respect to the understanding of the different advantages of intravenous (IV) to oral switch of antibiotics. The pooled analysis of the knowledge domain of the study questionnaire is summarized in Table 2.

### 3.3. Attitude Domain of the Questionnaire

Most of the participants (418; 76.8%) believed that their antibiotic prescribing behavior has an impact on the development of antibiotic resistance in their region. A relatively good proportion felt a combined role of all the stakeholders (doctors, patients, nurses, chemists, government, community) (313; 57.5%) in addressing the issue of antibiotic resistance (Figure 1).

The perceptions of respondents regarding the strategies that can be helpful in handling the issue of AMR are represented in Figure 2.

Table 3 illustrates the level of agreement among respondents in other antibiotic use practices. A relatively large number (478; 87.8%) agreed on the need for regular surveillance to combat AMR. Most of them disagreed on prescribing antibiotics on patients’ demands (488; 89.7%).

### 3.4. Practices Domain of the Questionnaire

The item on factors determining the choice of antibiotics prescribed revealed that culture susceptibility report was the most common factor (78.3%), followed by local resistance patterns (48%), cost of antibiotics (41.5%), recommendations from seniors/colleagues (27%), and own experience (26.6%) (Figure 3).

Treatment guidelines (62.3%) were the most commonly reported source of information used while making decisions on antibiotic prescriptions, followed by journals/textbooks (47%), internet/social media (40.8%), conferences/training/continuing medical education (CMEs) (32.7%) and expert opinion (28.5%). 172 (31.6%) participants agreed on employing delayed/back-up antibiotic prescribing in their clinical practice, while 222 (40.8%) disagreed, and 150 (27.6%) were not aware of the concept per se. Table 4 details the respondents’ responses to other questions in the practice domain.

203 (37.3%) participants had attended some training/conferences to update their knowledge of antibiotic use during the last 12 months.

### 3.5. Descriptive Statistics of KAP Scores

Table 5 summarizes the descriptive analysis of knowledge, attitude, and practices score among healthcare professionals in India. The majority of the participants had good or average scores in all the three domains studied.

### 3.6. Predictors of Aggregate KAP Score

Table 6 depicts the results of logistic regression analysis of the predictors of aggregate KAP score among the participating physicians. Specialists/super-specialists from basic and medicine/allied sciences were found to be associated with higher scores in comparison to non-specialists. Working in secondary healthcare settings was significantly associated with lower scores as compared to tertiary care. Physicians from the central zone were found to have significantly higher aggregate scores. Other factors such as age, gender, years of practice, and highest educational qualification were not found to have an influence on aggregate KAP scores among participants.

## 4. Discussion

The present cross-sectional survey was conducted with a viewpoint to evaluate the knowledge, attitudes, and practices in regard to antibiotic prescribing of physicians working in government as well as private settings across India with an ultimate goal of identifying knowledge gaps and informing interventions that could lead to judicious use of antimicrobials and help in containment of AMR.

Pooled analysis revealed that the majority of the participants had postgraduate qualifications and worked in tertiary care government settings. The respondents largely opined that indiscriminate use of antibiotics and use of broad-spectrum agents lead to the emergence of AMR, an observation in congruence with other similar surveys from India and other LMICs [18,19,20,25,26,27]. Most of the participants (478/544; 87.8%) agreed on regular surveillance of antibiotic use and resistance at local, regional, national, and global levels to combat AMR. Lack of access to essential antimicrobials leads to resorting to alternative agents, which may be less efficacious, in turn promoting the emergence and spread of resistance among pathogens [31], a fact which was though not appreciated by majority of participants. Furthermore, many respondents were not familiar with WHO AWaRe (access, watch and reserve) classification; thereby highlighting the need to educate physicians about it and the fact that as per WHO recommendation, more than 60 percent of antibiotic use in hospitals should be from “access” group [13]. Many studies conducted within and outside India have reported excess use of the “watch” group of antibiotics in hospital and community settings; a serious concern contributing to AMR [32,33,34,35,36]. Increasing awareness among physicians and sensitizing them towards favorable higher use of access compared to “watch” group of antibiotics is a crucial step towards optimal use of these agents. A relatively poor understanding of the concept of antibiograms was also evident in the study participants, with only 38% correct responses regarding the similarity of antibiograms for different hospitals, thereby suggesting the need for training in this area as well.

Restricted use of double anaerobic cover and intravenous (IV) to oral switch of antibiotics are among the key strategies defined for the successful implementation of antibiotic stewardship. We tried to identify the knowledge gaps, if any, pertaining to these concepts among study participants. Responses to the item on agent/s effective against anaerobes were quite dissatisfactory, with around 70 percent of respondents not being aware of anaerobic cover being provided by all the three agents (metronidazole, meropenem, and amoxyclav) listed in the questionnaire; lack of knowledge in this area probably leads to use of double anaerobic cover by physicians as reported in literature [35,36,37]. Most of the respondents were not able to appreciate all the listed benefits of switching intravenous antibiotics to the oral route when clinically desirable, a finding consistent with a survey in Nigeria where more than half of the participating physicians believed that parenteral antibiotics are more effective than oral ones [19]. Specific recommendations to sensitize physicians on intravenous to oral switch of antibiotics, and restricting the use of double anaerobic cover and use of multiple antibiotics effective against Gram-negative bacteria may be stressed upon.

WHO gave the slogan ‘Fight antibiotic resistance, it’s in your hands’ in 2017 to emphasize the crucial role of hand hygiene in preventing AMR [19]. Further, on 5 May 2022, i.e., World Hand Hygiene Day, WHO adopted the slogan ‘Unite for safety—clean your hands’ to prioritize the importance of hand hygiene [38]. Despite such measures at the international level, there is still a wide knowledge gap regarding the link between infection prevention and control (IPC) practices and AMR, as observed in our survey where only 53.8% of the participants believed that infection control measures (such as hand hygiene, cohorting, etc.) may help in handling the issue of AMR. In a similar earlier survey from India, less than 50% of respondents were reported to perceive inadequate hospital infection control as the cause of AMR [39]. In a survey in Nigeria, however, 88.5% of physicians reported following adequate hand hygiene practices [20]. Our findings are consistent with some other international surveys reflecting physicians’ under-rating of the role of IPC measures, including hand hygiene, in preventing AMR [19,40,41,42]. Hence, measures need to be taken to make physicians cognizant of the importance of various IPC practices in reducing the transmission of AMR in hospital settings.

A shorter duration of antibiotic therapy is another strategy defined under antimicrobial stewardship interventions. The basis for such recommendation is the presence of evidence that it is as efficacious as a longer duration and offers the added advantages of lower incidence of adverse effects and emergence of AMR [43,44,45,46,47]. However, findings from our survey with only 34.5% perceiving the importance of shorter duration of antibiotic therapy in limiting AMR call attention to conduct regular training and CMEs to impart updated knowledge regarding antibiotic prescribing to clinicians.

A relatively higher proportion of our study respondents (76.8%) believed that their prescribing practices may influence the development of AMR in their region compared to an earlier survey in an LMIC (50.3%) [20]. A good proportion also agreed on reducing non-prescription sales of antibiotics and limiting their use to cases with confirmed bacterial infections as key measures to handle the issue of AMR. The majority of the clinicians reported culture susceptibility report (78.3%) as the top criteria for choosing an antibiotic, followed by local resistance patterns (48%) and cost of antibiotics (41.5%). However, the proportion of respondents practicing a change/discontinuation of empiric therapy on the basis of the culture sensitivity report was not quite significant, reflecting dissonance between knowledge and practices. In contrast to an earlier survey conducted in West Bengal by Nair et al. [25], very few participants in our study (9/544;1.6%) depended on recommendations from pharmaceutical companies for deciding which antibiotics to prescribe. Treatment guidelines (62.3%) were the most commonly reported source of information used by study participants while making decisions on antibiotic prescription, an observation in agreement with most other similar surveys [19,20,22,25].

A sizeable portion of the respondents (392/544, 72%) preferred prescribing two or more classes of antibiotics in combination over single agents. A deeper analysis of the type and rationality of antibiotic combinations prescribed is demanded, which was, however, outside the scope of the present study. Future studies may also be planned in this field to assess the problem of irrational prescribing, if any, of antibiotic combinations.

Participants belonging to basic medicine and allied sciences had overall KAP scores higher than their surgical counterparts, a finding similar to an earlier survey in India [39]. Another cross-sectional study in north India reported a significant correlation between physicians’ experience in years and their perceptions regarding the importance of hand hygiene and culture susceptibility and the reported treatment of bacterial infection [27]. A survey conducted in Jordan by Karasneh et al. also reported significantly higher knowledge scores among physicians, specialists, and those having more years of experience [18]. Another survey conducted across primary care physicians in Quebec, however, reported that physicians with longer practice years were more likely to give in to patient pressures and prescribe antibiotics inappropriately [48]. Respondents from the central zone, mostly belonging to medical disciplines, had significantly higher scores compared to those from other zones of the country, a finding which, however, fails to draw any conclusions due to the very limited number of responses from this zone.

In the context of AMR and antibiotic prescribing practices, ours is the most comprehensive study involving physicians from all the zones of India, working in government as well as private settings, belonging to different specialties, with educational qualifications ranging from graduation to super-specialization and having a wide range of clinical experience. Hence, our results are more generalizable than earlier similar surveys from India [22,25,26,27,49,50,51].

The study had, however, several limitations due to resource constraints. The fact that we used convenience and snowball sampling approaches in the survey design must be taken into consideration while interpretating results due to associated drawbacks, such as selection bias and limited generalizability. In fact, the majority of the respondents (83.8%) belonged to tertiary care hospitals, and a very small proportion were from primary care hospitals (4.8%). Moreover, there were relatively unequal zone-wise participation rates with quite lesser representation from East and Central zones. Besides these concerns, shortcomings of self-reported data, such as giving desirable answers rather than true practices, undermining the credibility of results cannot be ignored. A useful approach to eliminate such social desirability bias could have been cross-checking the responses with actual prescription practices, which was, however, not feasible due to the online nature of the survey. We focused on the KAP of clinicians paying no heed to informal healthcare providers who are not qualified medical doctors but dispense antibiotics as part of their regular practice. Being cross-sectional in nature, the survey provided a snapshot of the views of participants, yet a follow-up study on the same participants may provide insights on changes in their attitudes over time, especially after the implementation of various strategies suggested here. Given the extensive data from across the country, we believe the study adds value in understanding the knowledge gap regarding optimal use of antibiotics among clinicians across the country and may help inform future strategies to improve antibiotic prescribing practices with the ultimate goal of fighting the battle against the global pandemic threat of AMR.

## 5. Conclusions

Our study findings have strengthened the case for regular conduct of continuing medical education and training of clinicians on various aspects related to antimicrobial resistance, surveillance, and use. On the basis of knowledge gaps identified in our survey, a few areas deserving attention are: antibiograms along with their interpretation and applicability in appropriate agent selection, WHO AWaRe classification of antibiotics, guideline-based recommendations for optimal use of antibiotic agents and duration of antibiotic therapy, double anaerobic cover and double cover for gram negative infections, irrational antibiotic combinations, and intravenous to oral switch of antibiotics when clinically desirable. Besides these, there is a need to emphasize the crucial role of infection prevention and control measures, including hand hygiene, not only among healthcare professionals but the community as a whole. Increasing awareness on AMR and sensitization and training of clinicians on various listed issues is a key strategy towards safe and rational use of antimicrobials and heralds the menace of AMR at local, national, as well as global, levels.

## Figures and Tables

**Figure 1 antibiotics-12-01496-f001:**
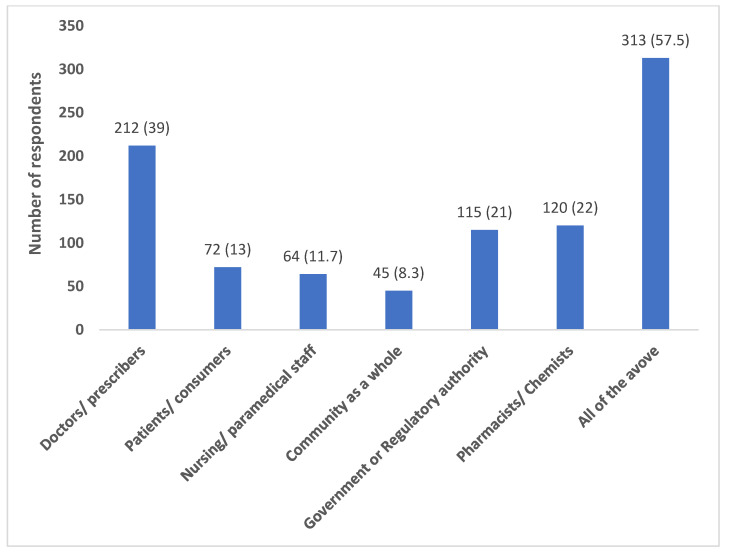
Respondents’ attitude regarding the stakeholders who can play key role in addressing the issue of antibiotic resistance. Data represented as numbers (percentages).

**Figure 2 antibiotics-12-01496-f002:**
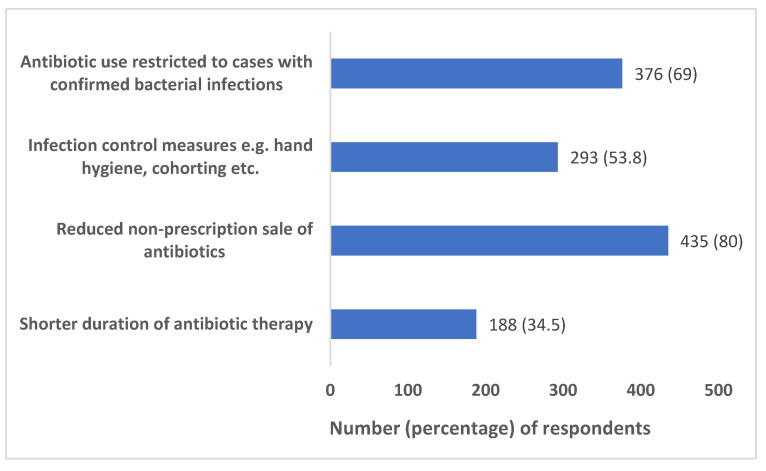
Perceptions of respondents regarding the strategies that can be helpful in handling the issue of AMR.

**Figure 3 antibiotics-12-01496-f003:**
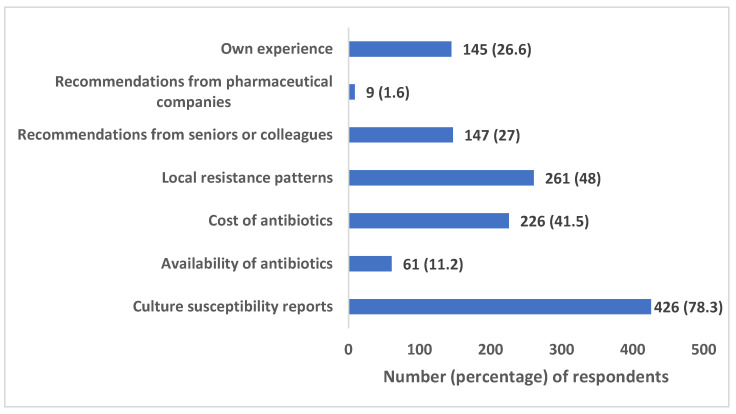
Factors determining the choice of antibiotics prescribed.

**Table 1 antibiotics-12-01496-t001:** Demographic profile of the study participants (N = 544).

Age in Years (Mean ± SD)	34.7 ± 10.3
Males, *n* (%) Females, *n* (%)	327 (60) 217 (40)
**Educational qualification**	
MBBS	180 (33.1)
MD/MS/Diploma/DNB	350 (64.3)
DM/MCh	14 (2.6)
**Specialty/Discipline**	
Basic sciences	173 (31.8)
Medicine and allied	153 (28)
Surgery and allied	130 (23.9)
Others (non-specialists)	88 (16.2)
**Zone-wise distribution**	
North zone	195 (35.8)
South zone	93 (17.1)
West zone	196 (36.0)
East zone (including Northeast)	37 (6.8)
Central zone	23 (4.2)
**Type of work setting**	
Government	432 (79.4)
Private	112 (20.6)
**Level of setting**	
Primary care hospitals (PHC)	26 (4.8)
Secondary care hospitals (CHC, District hospitals)	62 (11.4)
Tertiary care hospitals	456 (83.8)
**Years of practice**	
<5 years	300 (55)
5–10 years	95 (17.4)
11–20 years	86 (15.8)
>20 years	63 (11.6)

Data represented as *n* (%); MBBS: Bachelor of Medicine, Bachelor of Surgery; MD: Doctor of Medicine; MS: Master of Surgery; DNB: Diplomate of National Board; DM: Doctor of Medicine; MCh: Master of Chirurgiae; PHC: Primary health center; CHC: Community health center.

**Table 2 antibiotics-12-01496-t002:** Pooled analysis of the knowledge domain of study questionnaire (N = 544).

Question	Responses, *n* (%)
Q1. Indiscriminate use of antibiotics in humans, plants, and animals leads to antimicrobial resistance.	
YES	534 (98.2)
NO	10 (1.8)
Q2. Approximately 30 percent of all hospitalized patients receive antibiotics at any given time.	
YES	480 (88.2)
NO	12 (2.2)
NOT SURE	52 (9.5)
Q3. Lack of rapid diagnostic tests is one of the reasons for irrational antibiotic use.	
YES	432 (79.4)
NO	82 (15.1)
DON’T KNOW	30(5.5)
Q4. Limited access to essential antibiotics contributes to irrational antibiotic use and emergence of antibiotic resistance.	
YES	278 (51)
NO	126 (23)
MAYBE	140 (26)
Q5. Broad spectrum antibiotics, when used inappropriately, lead to emergence of antibiotic resistance.	
YES	521 (95.8)
NO	13 (2.4)
DON’T KNOW	10 (1.8)
Q6. Are you familiar with the WHO AWaRe classification of antibiotics?	
YES	245 (45)
NO	299 (55)
Q7. Antibiograms for different hospitals in a region are usually similar.	
YES	166 (30.5)
NO	207 (38)
DON’T KNOW	171 (31.5)
Q8. Hospital antibiograms serve as important tools in guiding empiric antibiotic therapy and tracking resistance patterns.	
YES	466 (85.7)
NO	12 (2.2)
DON’T KNOW	66 (12.1)
Q9. As per your knowledge, which is the most prescribed antibiotic in COVID-19 pandemic?	
Azithromycin	500 (91.9)
Doxycycline	218 (40)
Co-amoxyclav	70 (12.8)
Cefixime	37 (6.8)
Any other, please specify *	
Q10. Which of the following agents are effective against infections by anaerobes?	
Metronidazole	337 (61.9)
Meropenem	132 (24.2)
Co-amoxyclav	37 (6.8)
All of the above	171 (31.5)
Q11. When clinically desirable, the route of administration of antibiotics may be switched from intravenous to oral (IV to oral switch) due to the following reason/s:	
Decreases the duration of hospitalization	366 (67.3)
More convenient for patient	421 (77.4)
Increases the duration of hospitalization	0
Lesser complications	221 (40.6)

* Other antibiotics prescribed during COVID-19: ceftriaxone, gentamicin, piperacillin-tazobactam, cefuroxime, hydroxychloroquine, ivermectin. The correct responses to the questions have been printed in bold fonts.

**Table 3 antibiotics-12-01496-t003:** Attitude of respondents with regard to antibiotic use practices and surveillance. Data represented as numbers (percentages).

Attitude Based Questions	Strongly Agree	Agree	Neither Agree nor Disagree	Disagree	Strongly Disagree
There is a rampant use of antibiotics in the hospital I work in.	45 (8.3)	161 (29.6)	192 (35.3)	116 (21)	30 (5.5)
Surveillance of antibiotic use and resistance should be done regularly at hospital, local, regional, national, and global levels to combat antimicrobial resistance.	322 (59.2)	156 (28.7)	24 (4.4)	7 (1.3)	35 (6.4)
How much do you agree to prescribing antibiotics on patients’ demands?	1 (0.2)	14 (2.6)	41 (7.5)	192 (35.3)	296 (54.4)

**Table 4 antibiotics-12-01496-t004:** Participants’ responses to practice-based questions.

Question	Never	Rarely	Sometimes	Always
How frequently do you counsel the patients regarding appropriate use of antibiotics to prevent emergence of resistance?	12 (2.2)	43 (7.9)	185 (34)	304 (56)
How often do you change the empiric antibiotic prescribed on the basis of culture sensitivity report?	16 (2.9)	22 (4)	155 (28.5)	351 (64.5)
How often do you discontinue the empiric antibiotic in case of negative culture report?	41 (7.5)	55 (10.1)	223 (41)	225 (41.3)
How much do you prefer prescribing two or more class/es of antibiotics in combination over single agents?	27 (4.9)	125 (23)	346 (63.6)	46 (8.4)
How often do you prescribe antibiotic/s prophylactically without evidence of infection?	109 (20)	167 (30.7)	237 (43.6)	31 (5.7)

Data presented as numbers (percentages).

**Table 5 antibiotics-12-01496-t005:** Descriptive analysis of knowledge, attitude, and practice score among healthcare professionals in India.

Domain	Overall Score (Range)	Mean (SD) Score	Good Score; *n* (%)	Average Score; *n* (%)	Poor Score; *n* (%)
Knowledge	0–12	8 (1.6)	208 (38.2)	300 (55)	36 (6.6)
Attitude	5–27	20.2 (3.5)	287 (52.7)	243 (44.7)	14 (2.6)
Practices	5–21	15.3 (2.1)	278 (51)	262 (48.2)	4 (0.7)

**Table 6 antibiotics-12-01496-t006:** Logistic regression analysis of the predictors of aggregate score among physicians in India.

Variable	Odds Ratio (95% C.I.)	*p* Value
** *Gender* **		
Male	Reference	
Female	0.91 (0.58, 1.41)	0.67
** *Age group* **		
>50 years	Reference	
<30 years	1.16 (0.25, 5.3)	0.84
31–50 years	1.23 (0.31, 4.88)	0.77
** *Highest educational qualification* **		
MBBS	Reference	
MD/MD/DNB/Diploma	1.07 (0.57, 2.02)	0.82
DM/MCh	1.42 (0.37, 5.48)	0.61
** *Specialty/Super-speciality* **		
Non-specialists	Reference	
Basic sciences	2.95 (1.21, 7.2)	0.0
Medicine and allied sciences	2.71 (1.09, 6.67)	0.03
Surgery and allied sciences	1.28 (0.47, 3.46)	0.62
** *Type of healthcare setting* **		
Tertiary	Reference	
Primary	0.65 (0.22, 1.92)	0.44
Secondary	0.4 (0.18, 0.88)	0.02
** *Years of practice* **		
>20 years	Reference	
<5 years	1.13 (0.31, 4.09)	0.85
5–10 years	3.19 (0.91, 11.13)	0.07
11–20 years	2.11 (0.61, 7.33)	0.24
** *Zone* **		
North	Reference	
West	0.79 (0.48, 1.33)	0.38
South	1.05 (0.57, 1.95)	0.86
East	1.11 (0.47, 2.67)	0.8
Central	3.75 (1.39, 10.12)	0.009
*Constant*	0.1 (0.03, 0.35)	0

## Data Availability

The data presented in this study are available on request from the corresponding author. The data are not publicly available due to privacy concerns.

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
