# Peer review of "Awareness Regarding Antimicrobial Resistance and Antibiotic Prescribing Behavior among Physicians: Results from a Nationwide Cross-Sectional Survey in India"

_antibiotics, 2023, doi:10.3390/antibiotics12101496_

Round 1

Reviewer 1 Report

This manuscript presents an interesting overview on antibiotic use and antibiotic resistance awareness in India. However, some aspects should be improved.

INTRODUCTION: 

- Add more details about WHO AWaRE classification.

- Revise some sentences as they are not clear, such as "Various factors governing the healthcare professionals’ prescribing behaviours and how these relate to knowledge and attitude regarding AMR need to be understood if successful strategies to contain AMR are to be designed and implemented at regional and national levels." and "Another study reported high scores in knowledge and attitude scores among physicians in West Bengal though poor performance in practices."

METHODS

- I suggest to add the Questionnaire as supplementary materials and add less details to "Study questionnaire" section in the manuscript.

- Why the question "as per your knowledge, which is the most prescribed antibiotic in COVID-19 pandemic" was not included for scoring?

RESULTS: I think that both Table and Figure for each section of the questionnaire may be repetitive. If you add the complete questionnaire with all questions as supplementary material, you can try to reduce Results section.

- Specify the acronym "IV" the first time you used it.

DISCUSSION

- Try to be more precise about WHO AWaRE classification; add details.

- Specify better what you meant with "gram negative cover of antibiotics".

- Revise this sentence as it is not clear: "Further, on 5th May 2022, World hand hygiene day, WHO, under its campaign ‘‘Save lives: Clean your hands’, adopted the slogan ‘Unite for safety - clean your hands’ to prioritize hand hygiene improvement."

- Specify the acronym CMEs the first time you used it.

Author Response

Dear Sir,

Regards

Reviewer 2 Report

The study was carefully prepared and verified; the design can serve as an example for writing similar papers. However, composing the questions in the 'Knowledge' chapter can be a 'slippery field', and the interpretation of the answers can be misleading. Namely, some questions have an obvious answer, and it is difficult to make a mistake. The real question is how to evaluate this part of the questionnaire. More specific questions would be better for knowledge assessment, for example, relating the uncritical use of fluoroquinolones with developing resistance to carbapenems.

One study from Canada (PMID: 17923655) showed that older doctors are more inclined to give in to patient pressure and irrationally prescribe antibiotics, which could be commented on in the discussion. These results may be influenced by whether they were obtained before or after the publication of the WHO AWaRe recommendations, as well as whether these recommendations are essentially understood.

Author Response

Dear Sir,

Regards.
